# Protective Efficacy of Novel Oral Biofilm Vaccines against *Lactococcus garvieae* Infection in Mullet, *Mugil cephalus*

**DOI:** 10.3390/vaccines9080844

**Published:** 2021-08-01

**Authors:** Feng-Jie Su, Meei-Mei Chen

**Affiliations:** Department of Veterinary Medicine, National Taiwan University, No. 1, Section 4, Roosevelt Road, Taipei 10617, Taiwan; d06629003@ntu.edu.tw

**Keywords:** oral biofilm vaccine, *Lactococcus garvieae*, *Mugil cephalus*, IgM

## Abstract

*Lactococcus garvieae* (*L*. *garvieae*) is an important pathogen that causes enormous economic losses in both marine and freshwater aquaculture. At present, antibiotics are the only option for farmers to reduce the losses caused by *L. garvieae*. However, the usage of antibiotics leads to environmental pollution and the production of drug-resistant strains of bacteria. Therefore, vaccination is preferred as an alternative method to prevent infectious diseases. In this study, we describe an effective approach to the production of an oral biofilm vaccine, using bacteria grown on chitosan particles to form biofilms, and thus providing an inactive pathogen that enhances the immune response in fish. We observed the formation of a biofilm on chitosan particles and administered the novel oral biofilm vaccine to fish. We analyzed the immune responses, including antibody production, phagocytic ability, albumin/globulin ratio and immune-related genes, of vaccinated and control groups of black mullet. Our results show that the phagocytic ability of the biofilm vaccine group was 84%, which is significantly higher than that of the control group, and the antibody production in this group was significantly higher compared with the other group. The mRNA expression levels of immune-related genes (TLR2, IL-1β, TNF-α) were significantly upregulated in the spleen after vaccination. In challenge experiments, the relative percent survival (RPS) was 77% in the biofilm vaccine group, 18% in the whole-cell vaccine group, and 0% in the chitosan particle group at 32 days post-vaccination. In addition, we also found that the relative percent survival (RPS) at 1 day post-vaccination was 74% in the biofilm vaccine group, 42% in the whole-cell vaccine group, and 26% in the chitosan particle group. In both long-term and short-term challenge experiments, the viability of the biofilm vaccine group was significantly higher than that of the whole-cell, chitosan particle and PBS groups. We conclude that based on its protective effect, the *L. garvieae* biofilm vaccine is better than the whole-cell vaccine when challenged several weeks after vaccination. In addition, the biofilm vaccine also has a greater protective effect than the whole-cell vaccine when challenged immediately after vaccination. Therefore, the biofilm vaccine might represent a novel method for the prevention and treatment of *L. garvieae* infection.

## 1. Introduction

*Lactococcus garvieae* is a facultative anaerobic, non-motile and Gram-positive bacteria that infects many fish species and causes enormous economic losses in both marine and freshwater aquacultures [1]. In tilapia and grey mullet aquaculture, infection with *L. garvieae* leads to 70–100% morbidity rates, with clinical symptoms of exophthalmia, ascites haemorrhage and septicaemia [2,3]. In addition, surviving fish may suffer from chronic or persistent infections. It has been reported that biofilm formation is an important mechanism of chronic infection. Approximately 80% of persistent bacterial infections may be related to biofilm formation [4]. Biofilms are formed by bacteria, enabling them to resist environmental stresses such as oxidative stress, pH change, antibacterial substances, and the host immune system [5]. The composition of the biofilm matrix enables bacteria to deceive the host immune system and escape the immune response. For instance, after *Streptococcus pneumonia* infection, the bacteria trigger an acute response, as bacteraemia stimulates the immune system of the host and induces humoral immunity against planktonic bacterial antigens. Then, the bacteria form biofilms in the brain, spleen, kidneys and other tissues to escape the immune response. A bacterial biofilm consists of bacteria embedded in a complex array of extracellular substances, including extracellular DNA, peptidoglycans, extracellular proteins and a capsular layer [6]. Therefore, the biofilm antigenicity and immunogenicity are considerably different from those of the planktonic bacteria, and the biofilm elicits a different immune response in the host [7]. In previous studies, the different antigenicity and immunogenicity qualities of biofilms were identified and used to search for new antigens to improve existing vaccines or to develop new ones [8]. For instance, immunisation with polysaccharides from the biofilm matrix of *S. aureus* induced protective immunity against infections of the mammary gland of sheep and cows and prevented biofilm formation [9]. In a mouse model study, a biofilm LytB protein vaccine was immunogenic, and enhanced complement-mediated immunity and the phagocytosis of different serotypes of *S. pneumoniae* [10].

The development of a vaccine for oral administration involves confirming that the antigen is efficiently delivered to gut-associated lymphoid tissues (GALTs). GALTs are important immune tissues that play a crucial role in the gut to prevent infection [11]. Biofilms with multiple immunogenicity can protect bacteria from destruction by gastric acid. The biofilm vaccine model was further strengthened by the demonstration of the localisation and distribution of antigen in larger quantities for a longer duration in the gut and lymphoid tissues following oral vaccination in [12]. An oral biofilm vaccine was shown to induce specific IgM in vaccinated fish [13]. In addition, oral vaccines resulted in the upregulation of genes related to the recruitment of immune cells in [14]. These results suggest that biofilm is the best choice as the basis for the development of an oral vaccine.

Recent methods for developing oral biofilm vaccines mainly use chitin flakes, and have achieved good protective effects in a variety of fish species, such as common carp (*Cyprinus carpio*) [12,15], catfish (*Clarias batrachus*) [16], rohu (*Labeo rohita*) [17], red hybrid tilapia (*Oreochromis* sp.) [18] and Asian seabass (*Lates calcarifer*) [19]. In addition, oral vaccines use chitosan polymer-based, coated and inactivated bacteria to enhance protective efficacy [20,21].

In this study, we developed a new method for preparing an oral biofilm vaccine. We grew bacteria in suspension culture on chitosan particles to form a biofilm, and then the biofilm formation was monitored at different time points. Subsequently, we investigated the potential of the biofilm vaccine against *Lactococcus garvieae* by incorporating it into feed. We used immunohistochemistry (IHC) to monitor the delivery of oral vaccines. The antibody responses in the serum, the phagocytic ability, the albumin/globulin ratio and immune-related genes (C3, TLR2, IL-1β, and TNF-α) were analysed, and a challenge test was performed to confirm the protective efficacy of the oral vaccine.

## 2. Materials and Methods

### 2.1. Ethical Considerations

The present animal experimental study was approved by the Centre for Research Animal Care and observed by the Animal Care Use Committee of the National Taiwan University (protocol no. B201800003).

### 2.2. Experimental Fish

Black mullets were purchased from a mullet farm in Hsinchu, Taiwan. Two hundred and twenty-four black mullets that were maintained in an 800 L fibre-reinforced plastic (FRP) tank in brackish water (20 ppt) at 25 °C. For laboratory trials, fish (5 g ± 0.3 g) were reared in an indoor recirculated aquatic animal culture system at a regulated temperature of 26 ± 2.0 °C at the Department of Veterinary Medicine, National Taiwan University (Taipei, Taiwan). Tanks were provided with UV-treated and filtered water throughout this period, and 50% of the water was changed twice weekly. Spleens of 5 randomly selected fish were collected for bacterial isolation and polymerase chain reaction (PCR) analysis to confirm they were free from *L. garvieae* before conducting the experiments [22].

#### 2.2.1. Chitosan Particles

The method for preparing chitosan particles was modified from a previous study [23]. The chitosan particles were prepared by dissolving 1 g of chitosan powder (Sigma-Aldrich, Taiwan) in 1000 mL of ddH_2_O with 2% acetic acid, and then pH was adjusted to 7.0 with 1 N NaOH to form particles. The particles were stored at 4 °C.

#### 2.2.2. Bacterial Strains

*L. garvieae* was isolated from outbreaks in moribund black mullets at Taiwan fish farms. *L. garvieae* was isolated on blood agar (Oxoid^TW^, Creative Media Plate, Taiwan) and identified by sequencing 16S rRNA genes [22]. A single colony was isolated and enriched in 3 mL of brain–heart infusion (BHI; HiMedia, Creative Media Plate, Taiwan) broth containing 1% NaCl at 28 °C overnight in an incubator shaker. Then, cells were harvested by centrifugation at 10,000 rpm for 10 min. Subsequently, the pellet was resuspended in BHI with 18% glycerol as the stock suspension and stored at −20 °C.

#### 2.2.3. Cultured and Quantified *L. garvieae* Biofilm

The biofilm was quantified by the DMMB method, and the data were converted to colony-forming units. *L. garvieae* was cultured at a density of 10^4^ CFU/mL in 100 mL of BHI with 1% NaCl and 10 mg/mL chitosan particles and rotated at 100 rpm, and 1 mL samples were taken at 1, 3, 5, 7, 24, 48 and 72 h. The samples were washed twice with 0.01 M phosphate-buffered saline (PBS, pH 7.4) and harvested by centrifugation at 300 rpm for 10 min to remove excess medium on the chitosan particles. Then, 1 mL of PBS was added to each Eppendorf tube, and the biofilm was homogenised to disassociate it from the chitosan particles. BHI agar plates were inoculated with the homogeneity biofilm and incubated at 28 °C for 24 h. CFU/mL values were then calculated. The biofilm was quantitated using the DMMB method [24]. *L. garvieae* biofilm samples were measured at 620 nm using a spectrophotometer (BioPhotometer, Eppendorf, Taiwan). Scanning electron microscopy (SEM) was used to ensure that the chitosan particles were encapsulated in the biofilm. The SEM samples were viewed with a JeoL JSM-7800F (JEOL Ltd., Tokyo, Japan) scanning electron microscope at 15 kV using a 15–17 mm working distance, and photographs were taken.

#### 2.2.4. Preparation of *L*. *garvieae* Biofilm and Whole-Cell Vaccines

*L. garvieae* bacteria were cultured in 100 mL of BHI broth containing 1% NaCl and 10 mg/mL chitosan particles and rotated (100 rpm) at 28 °C for 48 h. Then, biofilms were harvested by centrifugation at 300 rpm for 10 min at 4 °C and washed thrice with sterile PBS. In addition, we added 1 mL PBS with biofilm into Eppendorf tubes, and the biofilm was homogenised to disassociate it from the chitosan particles. BHI agar plates were inoculated with the homogeneity biofilm and incubated at 28 °C for 24 h. CFU/mL values were then calculated. The final biofilm was inactivated with 2% (*w/v*) formalin for 24 h. Inactivated biofilm was washed thrice with sterile PBS. The biofilm vaccine was stored at 4 °C. For the preparation of the whole-cell vaccine, the described strain was inoculated in BHI containing 1% NaCl and rotated at 100 rpm at 28 °C for 7 h. The pellet was harvested by centrifugation at 10,000 rpm for 10 min and washed thrice using PBS. In addition, 1 mL PBS with whole cells was added into Eppendorf tubes and grown on BHI agar plates at 28 °C for 24 h; then, CFU/mL values were calculated. The resulting cells were inactivated with 2% (*w/v*) formalin for 24 h. Inactivated whole cells were washed thrice with sterile PBS. The whole-cell vaccine was stored at 4 °C.

### 2.3. Preparation of Feed-Based Biofilm and Whole-Cell Vaccines

The feed-based method was modified from a previous study [18]. The biofilm vaccine and whole-cell vaccine were each resuspended in PBS solution to a concentration of 10^10^ CFU/mL and mixed thoroughly before spraying onto a pellet mixture using a homogeniser (PowerMasher II, OPTIMA Ltd., Tokyo, Japan) to incorporate the vaccines into the commercial feed. Finally, the prepared feed was pelleted using a pelletiser machine and dried at 70 °C for 24 h in an oven before the experiment.

### 2.4. Vaccination

#### 2.4.1. Experimental Design 1

The time course of vaccination and sampling are shown in Figure 1. After one month of acclimation in FRP tanks, the fish were divided into four groups and moved to 80 L FRP tanks, each containing 23 fish. Four groups were established: the biofilm vaccine group, whole-cell group, the chitosan particle group and the PBS group. Each were vaccinated for 14 consecutive days. The collected samples were analysed for immune-related genes. The collected blood was analysed to measure phagocytic ability, albumin–globulin ratio and antibodies after vaccination. The vaccinated fish in each group were challenged with *L. garvieae* at 1 day post-vaccination (Figure 1). Finally, the survival rate was calculated.

#### 2.4.2. Experimental Design 2

The time course of vaccination and sampling is shown in Figure 1. Experiment 2 was designed for the long-term evaluation of antibody titres and *L. garvieae* challenge tests. The fish were divided into four groups and moved to 80 L FRP tanks, each containing 33 fish. Four groups were established: the biofilm vaccine group, the whole-cell group, the chitosan particle group and the PBS group; each were vaccinated for 14 consecutive days. The blood was collected 7, 14, 21, 32, 70 and 100 days post-vaccination. Twenty-five fish in each group were randomly selected and challenged with *L. garvieae* at 32 days post-vaccination (Figure 1). Finally, the survival rate was calculated. The remaining 8 fish in each group were not challenged with the pathogen, and the antibody response was monitored on days 70 and 100 following the completion of the vaccination program.

### 2.5. Immunohistochemistry

The fish were euthanised by anaesthetic, and the abdominal cavity was cut to obtain the spleen, head kidney and intestine. The spleen, head kidney and intestine were fixed in 10% neutral buffered formalin for at least 24 h, and blank tissue sections were prepared for immunohistochemical staining. Tissue sections were deparaffinised and then rehydrated following a standard procedure. The rehydrated slides were immersed in Trilogy™ (Cell Marque, Rocklin, CA, USA) at 121 °C for 15 min for antigen retrieval. The tissue sections were cooled down on the bench top for 30 min. Then, the CRF Anti-Polyvalent HRP Polymer (DAB) stain kit (ScyTek, Taiwan) was applied according to the manufacturer’s instructions. To detect biofilm antigens, tissue sections were incubated with primary antibody (polyclonal rabbit anti-*L. garvieae* biofilm antibody from our laboratory) at a dilution of 1:500 in 0.05 M Tris buffer (pH 7.6) for 1 h. The antibody was replaced with non-immune rabbit serum in negative controls.

### 2.6. Phagocytic Ability and Albumin–Globulin Ratio Analyses

After vaccination, 3 fish from each experimental group were taken randomly and sedated with 30 ppm tricaine methanesulfonate (MS-222) in brackish water. Blood was drawn from the caudal vein using a 1 mL plastic syringe (27G) rinsed with 120 mM EDTA. Clean brackish water was used to observe the state of the fish, which were returned to the experimental group after they recovered and resumed swimming. For the phagocytic assay, 10^7^ cells of formalin-killed *L. garvieae* were added to 50 μL of pooled blood samples in a sterile microplate and incubated for 30 min at 25 °C after thoroughly mixing in the well. Following incubation, the blood–bacteria suspension was pipetted, and 25 μL of this suspension was smeared on 3 glass slides. After air drying, the smears were fixed in 95% ethanol, redried and stained with May–Grunwald–Giemsa. The phagocytic cells with phagocytosed bacteria were counted [25]. The albumin–globulin ratios (A/G) in the sera were calculated by following the method of Thanga Viji et al. [26]. The plasma samples were analysed for total protein (following the dye-binding method of Bradford (1976) using bovine serum albumin as the standard), albumin (using the bromocresol green method (MeDiPro, Taiwan)) and globulin (by subtracting the albumin value from the total protein value). Finally, the A/G ratio was calculated.

### 2.7. Antibody Detection

Serum antibodies were collected using a method modified from a previous study [18]. Three fish from each experimental group were taken randomly and sedated with 30 ppm tricaine methanesulfonate (MS-222) in brackish water. Blood was drawn from the caudal vein using a 1 mL plastic syringe (27G) without anticoagulant and stored at 4 °C overnight (to collect the serum), after which serum was extracted by centrifugation for 5 min. Clean brackish water was used to observe the state of the fish, which were returned to the experimental group after they recovered and resumed swimming. Antibody titres against *L. garvieae* biofilm or whole-cell bacteria were assayed by ELISA. Briefly, a 96-well microplate was coated with 1 × 10^8^ CFU/mL *L. garvieae* biofilm or whole-cell bacteria in 100 μL of coating buffer (50 mM carbonate buffer, pH 9.6) at 4 °C overnight, then washed with PBS containing 0.05% Tween-20 (PBST) and blocked in PBST containing 1% BSA (PBSTB) at 28 °C for 1 h. After blocking, the wells were washed three times with 150 μL of PBST, and mullet serum (100 μL/well) in a 1:300 dilution in PBST was added to the wells. After incubation at 28 °C for 1 h, they were washed three times, and 100 μL of mouse-anti-fish IgM HRP monoclonal antibody (Aquatic Diagnostics Ltd., Stirling, UK, 1:300 dilutions in PBST) was added to each well. Then, the 96-well microplate was incubated at 28 °C for 1 h and washed three times with 150 μL of PBST. PBST was removed, and 100 μL/well TMB was added. Finally, 50 μL of H_2_SO_4_ (2 M) was added after 10 min at 28 °C. Then, the microplate was read at 450 nm with a 96-well microplate reader as soon as possible.

### 2.8. RNA Isolation and Real-time Quantitative PCR (RT-qPCR)

The fish were euthanised by anaesthetic, and the abdominal cavity was cut to obtain the spleen. Total RNA was extracted from the spleens of vaccinated fish using an RNA kit (Geneaid Co., Ltd., Taiwan) according to the manufacturer’s instructions. The extracted RNA was eluted using RNase-free water, and RNA samples were aliquoted and stored at −80 °C until use. Single-stranded cDNA was synthesised from 1 μg of total RNA using a GoScript™ Reverse Transcriptase Kit (Promega Co., Ltd., Taiwan) following the manufacturer’s instructions. Real-time quantitation was performed in an ABI StepOnePlus Real-Time PCR System (Applied Biosystems Ltd., Foster City, CA, USA) using TOOLS SYBR Green qPCR Mix (ZEJU Co., Ltd., Taiwan) following the manufacturer’s recommendations. For quantitative real-time PCR, the specific primer pairs were designed as shown in Table 1 [27]. The EGF-1α gene was used as a housekeeping gene, and it was amplified using EGF-1α-F and EGF-1α-R gene-specific primers. All samples were run in triplicate, and each assay was repeated three times. After finishing the programme, the threshold cycle (Ct) values were obtained from each sample. Relative gene expression levels were evaluated using the 2^−ΔΔCT^ method.

### 2.9. Challenge with L. garvieae in Experiments 1 and 2

Fish were challenged by immersion with 10^7^ CFU/mL *L. garvieae* on day 1 or day 32 post-vaccination. The challenged fish were observed for the next 10 and 30 days post-challenge, and mortalities were recorded daily. The relative percentage survival (RPS) was calculated according to the following formula: RPS = (1 − % mortality of immunised group/% mortality of PBS group) × 100%.

### 2.10. Statistical Analysis

GraphPad Prism software (Version 8.0; GraphPad Software, Inc., San Diego, CA, USA) was used to generate graphs and perform statistical analyses. The results are presented as the mean ± SD of three replicates. The phagocytosis, A/G ratio, antibody response and relative mRNA expression were subjected to one-way ANOVA followed by tukey’s multiple comparison tests. A *p*-value < 0.05 was considered significant. A log-rank test was used to compare the difference in survival time between the biofilm vaccine group, the whole-cell group, the chitosan particle group and the PBS group.

## 3. Results

### 3.1. SEM Observation of Biofilm Formation

Biofilms were grown on chitosan particle surfaces and were quantitated at different time points. The results show that bacteria attached to the chitosan and began to form a biofilm at 3 h, reaching a peak at 48 h (Figure 2A,C). Since the correlation between biofilm DMMB and CFU was R = 0.97 at 48 h (data not shown), in this study we quantified the biofilm by CFU. Figure 2B shows chitosan particles without *L. garvieae* bacteria.

### 3.2. Confirmation of Adsorbed Biofilm Antigen in Intestine, Head Kidney and Spleen via Immunohistochemistry

Black mullets were orally administered the biofilm vaccine, and the antigen was detected by IHC. No signal was detected in the PBS group (Figure 3A,C,E), while the group that received the biofilm vaccine displayed signals in the intestine, head kidney and spleen tissues (Figure 3B,D,F).

### 3.3. Innate Immune Response after Vaccination

After the vaccines and control diets were orally administered to the black mullets, the immune responses of experimental fish were evaluated by measuring phagocytic ability and A/G ratios. The results indicate that the phagocytic ability of the biofilm vaccine group (84%) was significantly higher than that of the whole-cell vaccine, chitosan particle and PBS groups (66%, 65% and 59%, respectively) (Figure 4A). The A/G ratio was also increased in the biofilm vaccine group (0.56) compared to the other three groups (0.34, 0.42 and 0.32, respectively) (Figure 4B).

### 3.4. Antibody Production

In order to evaluate the antibody response against biofilm and whole-cell bacteria after vaccination, 10^7^ (CFU/well) whole cells or biofilms were coated in 96-well plates with serum and used to measure IgM titres. The anti-*L. garvieae* biofilm IgM titre in the biofilm vaccine group was significantly higher than those in PBS and whole-cell groups (*p* < 0.05) (Figure 5A). The OD450 value of the biofilm vaccine group was 0.22, and those of the chitosan particle, whole-cell vaccine and PBS groups were respectively 0.19, 0.18 and 0.18. In addition, the anti-*L. garvieae* whole-cell IgM titre in the biofilm vaccine group was significantly higher than those in the PBS and whole-cell groups (*p* < 0.05) (Figure 5B). The OD450 value in the biofilm vaccine group was 0.218, and those in the chitosan particle, whole-cell vaccine and PBS groups were 0.19, 0.195 and 0.184.

### 3.5. Immune-Related Gene Analysis

After vaccination, mRNA was extracted from the spleen and analysed for immune-related gene expression, including interleukin 1β (IL-1β), toll-like receptor 2 (TLR2), tumour necrosis factor-α (TNF-α) and complement component 3 (C3). The relative expression analysis showed that the mRNA expression levels of IL-1β, TLR2 and TNF-α genes were significantly increased (*p* < 0.05) in the biofilm vaccine group when compared to the PBS group. TNF-α expression in the biofilm vaccine group was 7.86 times higher than that in the PBS group, and its expression in the whole-cell vaccine and chitosan particle groups was 2.6 and 0.1 times higher than that in the PBS group (Figure 6A). The expression of the C3 gene was higher in the vaccine groups, but the difference was not significant (Figure 6B). IL-1β expression in the biofilm vaccine group was 3.4 times higher than that in the PBS group, and its expression in the whole-cell vaccine and chitosan particle groups was 1.6 and 1.1 times higher than that in the PBS group (Figure 6C). TLR2 expression in the biofilm vaccine group was 2.4 times higher than that in the PBS group, and its expression in the whole-cell vaccine and chitosan particle groups was 0.5 and 1.3 times higher than that in the PBS group (Figure 6D). Overall, the results show that pro-inflammatory-related gene expression in the oral biofilm vaccine group was significantly increased compared to that in the other groups.

### 3.6. Relative Percentage Survival

Black mullets were orally administered different vaccines daily for 14 days and then challenged with 10^7^ CFU/mL *L. garvieae*. The death of fish was observed, starting at 3 days after the challenge and increasing gradually. Nine days after the *L. garvieae* challenge, the survival rates for the groups administered feed with the biofilm vaccine, whole-cell vaccine, chitosan particles and PBS were respectively 75%, 45%, 30% and 5% (Figure 7), and the differences were significant. The relative percent survival (RPS) was 74%, 42% and 26% for the biofilm vaccine, whole-cell vaccine and chitosan particle groups, respectively (Table 2).

### 3.7. Long-Term Analysis of Serum Antibody Titres and Relative Percentage Survival

The biofilm vaccine group showed a significant increase (*p* < 0.01) in IgM titres compared to the other groups at 7, 14, 21, 32, 70 and 100 days post-vaccination. The IgM titre peak was observed at 14 days post-vaccination in the biofilm vaccine group and was significantly higher than those in the other groups on days 7, 14, 21, 32 and 70. The IgM titres of the chitosan group was slightly increased on days 7, 14, 21 and 32, but no significant changes were observed throughout the study period. The IgM titres of the whole-cell group slightly increased on days 7, 14, 21, 32 and 70, but no significant changes were observed throughout the study period. The PBS group showed no significant changes in antibody levels throughout the study period (Figure 8). The experimental fish were challenged with *L. garvieae* at 32 days post-vaccination. Survival rates in the biofilm vaccine, whole-cell vaccine, chitosan particle and PBS groups were 80%, 28%, 13% and 12% at 30 days after *L. garvieae* challenge, respectively, and the differences were significant (Figure 9). The RPS was 77%, 18% and 0% for the biofilm vaccine, whole-cell vaccine and chitosan particle groups, respectively (Table 3).

## 4. Discussion

*L. garvieae* causes high mortality in aquaculture fish in Taiwan, especially in summer. We believe that the recurrence of streptococcosis is due to biofilm formation, causing chronic infection [28]. Biofilm provides a stable environment for *L. garvieae*, from which it can spread to other tissues, resulting in reinfection. A previous study showed that whole-cell counterparts—that is, bacteria living in biofilms—have a developmental evolutionary advantage in comparison to planktonic cells because they are less sensitive to antibiotics, which complicates the effectiveness of antimicrobial therapy. Moreover, biofilm formation leads to inflammatory response inhibition, antibody neutralisation and macrophage phagocytosis [29,30,31]. In order to solve the problem of long-term chronic infections, researchers have recently begun to develop oral biofilm vaccines against a diverse range of biofilm antigens. However, although biofilm is easy to culture, it is difficult to mass produce for the preparation of vaccines. The earliest biofilm collection method used chitin flakes, but this procedure is complicated and time-consuming [15]. In this study, we used chitosan particles to provide an adherence platform for cultured bacterial biofilm in suspension. In addition to this function, chitosan is an immune stimulant and can be used as an adjuvant in fish [32]. In our experiments, we observed that *L. garvieae* biofilm formation was detectable on chitosan particles at 3 h. At 48 h, the largest amount of *L. garvieae* biofilm was observed on chitosan particles via SEM, and the observations were in agreement with the results of the DMMB method used to create the growth curves of biofilm (Figure 2A).

Oral administration is the simplest method for vaccination in aquaculture. It was previously revealed in salmon that oral vaccination can induce the production of antibodies with protective efficacy [33]. The oral administration of vaccines and adjuvants (e.g., oil-based and aluminium-based adjuvants) has been reported to induce mucosal immunity, particularly in the gut, and to achieve a good protective effect [34,35,36,37] because of the resistance to gastric acid [38,39,40]. *L. garvieae* survives at pH 2–9 and is resistant to extreme environments [41]. Therefore, when our oral biofilm vaccine passes through stomach acid, it is not destroyed and can be absorbed by intestinal epithelial cells. In our study, IHC staining detected the oral biofilm vaccine antigen in the intestine, head kidney and spleen. Strong signals were observed in the small intestine and spleen (Figure 3). This indicates that our biofilm vaccine can be effectively delivered to the immune tissues of the fish.

Phagocytic capacity and A/G ratio are important parameters of the innate immune response. Our results show that mullet fish are able to mount an immune response against *L. garvieae* bacteria after vaccination. The phagocytic ability in the biofilm group was 84%, and the A/G ratio was 0.56, which values are higher than those in the other groups. However, the phagocytic ability in the whole-cell group was 66% and the A/G ratio was 0.34, which values are not significantly higher than those in the other groups. In addition, the chitosan particle group had a phagocytic ability of 65% and an A/G ratio of 0.42. The same findings were reported in a previous study by Ranjan et al., who found that feeding chitosan increased the phagocytic capacity and A/G ratio of seabass [42]. Therefore, we hypothesise that chitosan particles might also act as an adjuvant for biofilm vaccines to enhance the innate immunity of mullet fish.

TLRs contribute to the recognition of pathogen-associated molecular patterns (PAMPs) and activate immune cells in response to infection through signalling pathways. TLR2, one member of the TLR family, has a highly conserved structure involved in detecting the cell walls of Gram-positive bacteria, and the TLR2 activation signalling pathway induces the expression of TLR2 and cytokines such as IL-1β and TNF-α [43,44]. In the relative mRNA expression study, TLR2, IL-1β and TNF-α mRNA were significantly increased in the oral biofilm vaccine group. The expression of TNF-α mRNA was significantly upregulated in the whole-cell vaccine group, but TLR2 gene expression was lower than that in the PBS, whole-cell and biofilm vaccine groups. The expression of TLR2 and IL-1β was increased, but not significantly, in the chitosan particle group. IL-1β has been shown to enhance antibody production when administered with bacterial vaccines, suggesting that it might be effectively exploited as an immune adjuvant to improve vaccine efficacy [45]. In *Mycobacterium marinum*-infected zebrafish, TNF-α was shown to promote macrophage phagocytosis and restrict bacterial growth in infected macrophages [46]. In the complement system (C3), although there was an average difference, it was not significant. According to Rao et al., the expression of IL-1β, TNF-α and C3 complement genes increased after the injection of a *Lactococcus garvieae* bacterial vaccine [47]. Although the methods and types of vaccines are different, the pro-inflammatory response also increased in this study. Therefore, the biofilm vaccine group induced a pro-inflammatory response through IL-1β and TNF-α gene expression via the TLR2 signal transduction pathway, enhanced the production of antibodies, and increased the phagocytic ability of macrophages.

Antibody production is an important index to evaluate vaccine efficacy. In previous oral vaccines research, when antigens were delivered via the gut, local and systemic immune responses were elicited, indicated by high amounts of circulating IgM [48]. In the present study, the results after biofilm vaccine administration revealed significantly increased IgM against biofilm antigen or whole-cell antigen (Figure 5). No significant antibody production was observed in the chitosan particle or whole-cell vaccine groups. Biofilm vaccines incorporate a variety of antigens, including whole cells. This also shows that oral biofilm vaccines can produce anti-whole-cell antibodies, but whole-cell vaccines may be present in insufficient amounts in the feed to produce more antibodies.

After vaccination, we monitored the production of fish antibodies for 100 days. The IgM level in the biofilm vaccine group was significantly different from that in the PBS group. Antibody production was observed on days 7, 14, 21, 32, 70 and 100. However, the chitosan particle and whole-cell vaccine groups did not display significant differences in the production of antibodies. We conclude that the biofilm vaccine stimulated long-term anti-*L. garvieae* antibody production an immune response that is transferred from GALT to systemic over a long period. In studies of *Pseudomonas putida* and *Streptococcus*
*agalactiae* biofilm vaccines, the same enhancement in antibody production was also found [18,49].

In challenge experiments 1 and 2, the RPS values were 74% and 77% for the biofilm vaccine group, 42% and 18% for the whole-cell vaccine group and 26% and 0% for the chitosan particle group, respectively. According to the cumulative survival rates in these two challenge experiments, the survival rates of whole-cell vaccine and chitosan particle groups decreased sharply, showing reduced long-term protective efficacy after bacteria challenge (Table 3) and suggesting limited applicability. Our results demonstrate that the oral administration of the biofilm vaccine triggered IgM antibody production, and the protective effect against the infection of *L. garvieae* lasted for at least 70 days post-vaccination. Most studies on biofilm vaccines have focused on vaccines against *A*. *hydrophila*. In a previous study on an *A*. *hydrophila* oral vaccine, fish were fed biofilm (BF) and free cell (FC) vaccines against *A. hydrophila* at 10^10^ cells/g fish/day for 20 days. Upon challenge with *A. hydrophila* at 10^9^ CFU/mL, BF-vaccinated fish had a significantly higher relative percent survival (88%) than the FC-treated fish (29.6%) at 60 days post-vaccination [50]. A recent study of an inactivated *L. garvieae* oral vaccine with chitosan–alginate capsules reported an RPS of 66.67 ± 5.77% compared to the control group [51].

## 5. Conclusions

In this study, we developed a novel, effective, orally administered chitosan particle biofilm vaccine, which conferred effective protection against *Lactococcus garvieae* in mullet fish. The *L. garvieae* oral biofilm vaccine significantly increased specific antibody titres, enhanced phagocytosis, and induced pro-inflammatory gene expression. Although the *L. garvieae* biofilm vaccine had favourable protective effects in grey mullet, the detailed mechanism of the oral vaccine is still not understood and should be explored in future investigations. This study provides a novel method for the further development of vaccines that are convenient and more applicable to the aquaculture industry.

## Figures and Tables

**Figure 1 vaccines-09-00844-f001:**
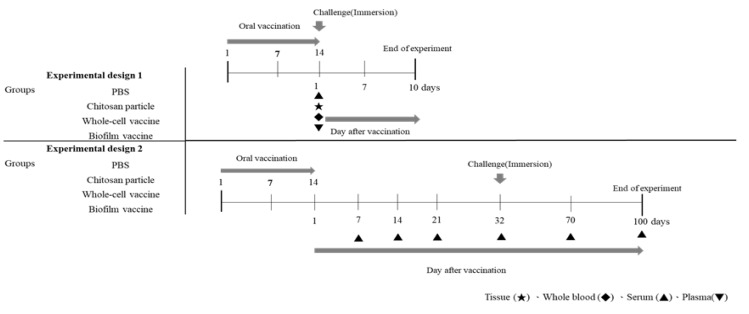
Graphical overview of the vaccination/challenge schedule and sampling time points.

**Figure 2 vaccines-09-00844-f002:**
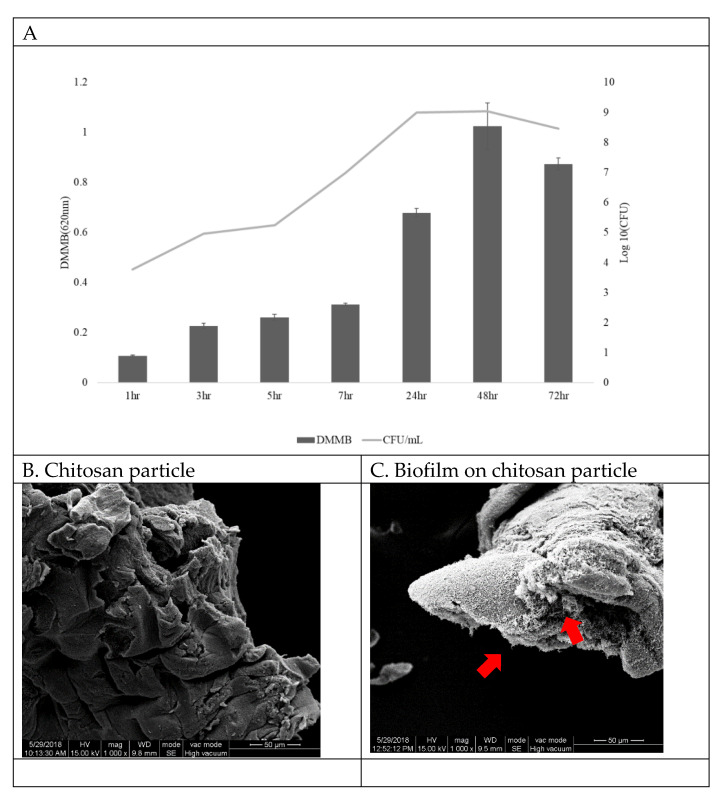
(**A**) The biofilm bacteria growth curve on chitosan particles (OD/h). *L. garvieae* was inoculated into BHI broth with 1% NaCl. The starting inoculum was 10^4^ CFU/mL. From the same sample, the biofilm growth curve was quantified by the DMMB method, and CFU values were plotted to obtain the bacteria growth curve. The biofilm growth abilities were evaluated by determining the optical density (OD) after applying the DMMB method (from time = 0 to time = 72 h). Data are presented as mean ± SD (*n* = 4). Scanning electron micrographs of (**B**) chitosan particle surface without *L. garvieae* growth and (**C**) biofilm growth on a chitosan particle at 48 h (arrows: *L. garvieae* biofilm). Magnification: 1000×; bars = 50 μm.

**Figure 3 vaccines-09-00844-f003:**
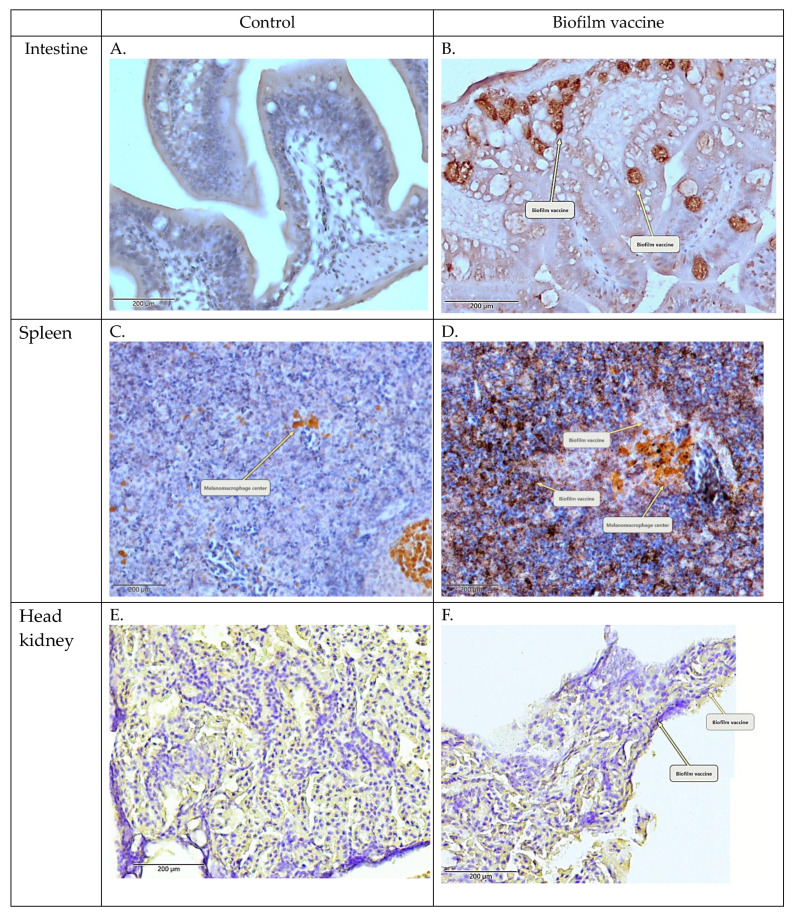
Immunohistochemical (IHC) staining of the head kidney and intestine collected after vaccination. (**A**,**C**,**E**) PBS control. (**B**,**D**,**F**) Biofilm vaccine group (mag. ×400). Arrows indicate positive signals and melanomacrophage centres.

**Figure 4 vaccines-09-00844-f004:**
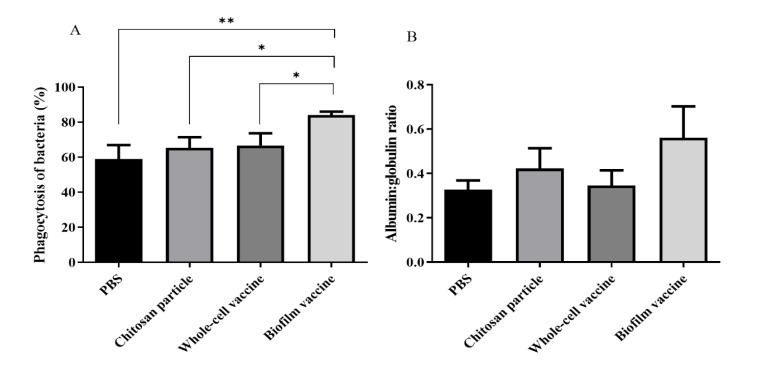
(**A**) Phagocytic activity of whole blood with different types of vaccine. The biofilm vaccine group had significantly increased phagocytic activity compared to PBS and whole-cell vaccine groups. Data are presented as mean ± SD (*n* = 3 fish × 3 replicates per treatment). *p*-values were calculated by one-way ANOVA (*p* < 0.01 **, *p* < 0.05 *). (**B**) Albumin–globulin ratios of plasma with different vaccines. Data are presented as mean ± SD (*n* = 3 fish × 3 replicates per treatment). The *p*-values indicate that there were no significant differences between groups.

**Figure 5 vaccines-09-00844-f005:**
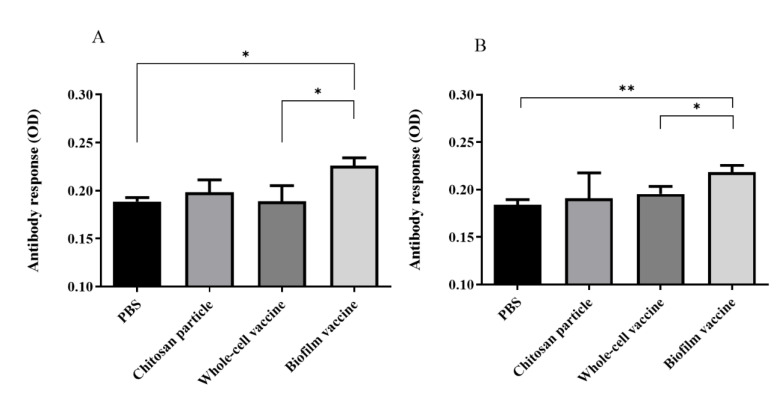
Serum IgM titres of fish vaccinated with different vaccines. The biofilm vaccine group had significantly increased IgM titres compared to the PBS and whole-cell vaccine groups. (**A**) Biofilm antigen coating on 96-well microplate. (**B**) Whole-cell antigen coating on 96-well microplate. Data are presented as mean ± SD (*n* = 3 fish × 3 replicates per treatment). *p*-values were calculated by one-way ANOVA (*p* < 0.01 **, *p* < 0.05 *).

**Figure 6 vaccines-09-00844-f006:**
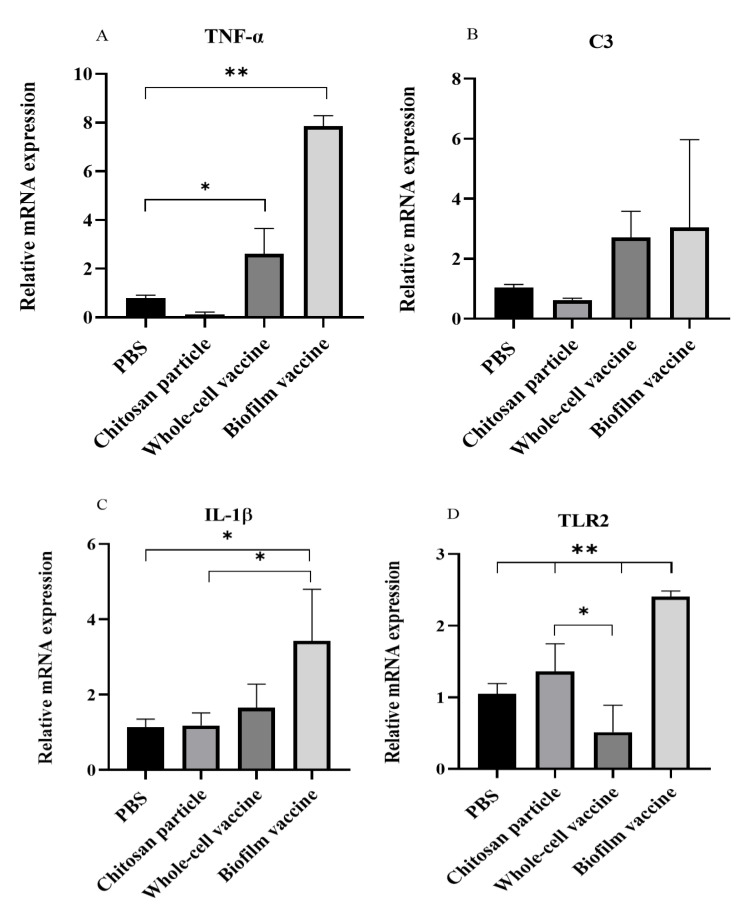
Expression levels of (**A**) TNF-α (**B**) C3, (**C**) IL-1β and (**D**) TLR2 in spleen samples (*n* = 3 fish spleen samples × 3 replicates per treatment) after vaccination in each group. Data are presented as mean ± SD, and *p*-values were calculated by one-way ANOVA (*p* < 0.05 *, *p* < 0.01 **).

**Figure 7 vaccines-09-00844-f007:**
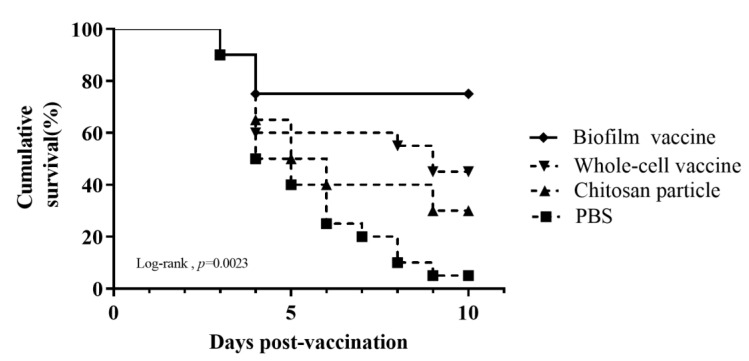
Cumulative survival rate in challenge experiment 1. The mullets were challenged by immersion with 10^7^ CFU/mL of *Lactococcus garvieae*. Challenge was performed on day 1 after vaccination. Fish were monitored for 10 days. Statistical significance was determined by the logrank test; there were significant differences between groups (*p* = 0.0023).

**Figure 8 vaccines-09-00844-f008:**
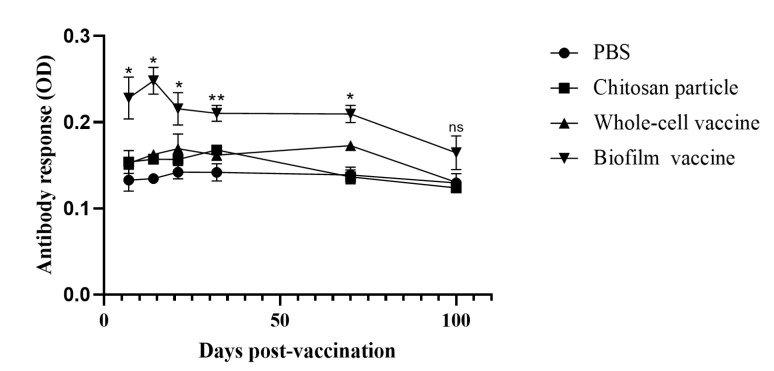
Long-term analysis of challenge experiment 2. Serum antibody levels of fish vaccinated with different vaccines. Data are presented as mean ± SD (*n* = 3 fish × 3 replicates per treatment). *p*-values were calculated by one-way ANOVA. Data are significantly different (*p* < 0.05 *, *p* < 0.01 **).

**Figure 9 vaccines-09-00844-f009:**
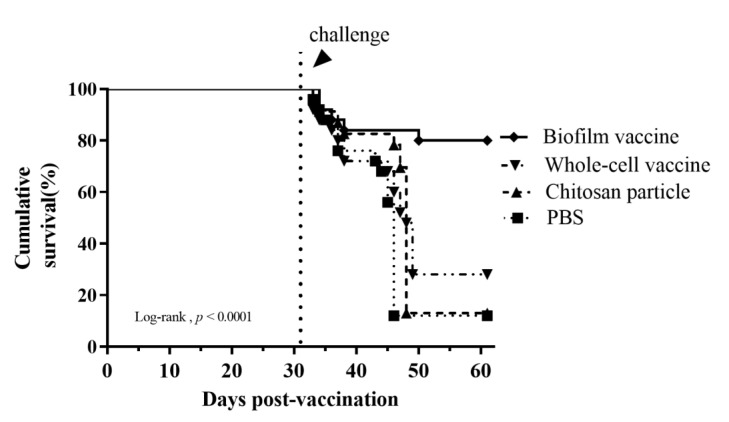
Long-term analysis of relative percentage survival in challenge experiment 2. Cumulative survival rate of the experimental mullet challenged by immersion with 10^7^ CFU/mL of *Lactococcus garvieae*. Challenge was performed on day 32 post-vaccination. Fish were monitored for 30 days. Statistical significance was determined by the log-rank test; there were significant differences between groups (*p* < 0.0001).

**Table 1 vaccines-09-00844-t001:** Primer name, sequence, target gene and application in the present study.

Name	Sequence	Tm (°C)	Reference
IL-1β-F	GAGGAGCTTGGTGCAGAACA	61.4	[27]
IL-1β-R	CTTTGTTCGTCACCTCCTCCA		
C3-F	GCATCACGCTCCTTGTCTTT	61.4	[27]
C3-R	ACCACTATGCCACAAGAACATC		
EGF1α-F	CTTCTTCTGGGCCTTCTCT	60	this paper
EGF1α-R	CTTGGACGTTTCGCTGTC		
TLR2-F	CTTTCTCCTCGTCCCTCTG	60	this paper
TLR2-R	CGTGTTTGTTGTGGTCT		
TNF-α-F	GCGCAGTCTGTCATTGGTT	60	[27]
TNF-α-R	ACTGGACACGCTCACTGTAGTG		

**Table 2 vaccines-09-00844-t002:** Relative percent survival (RPS) in vaccinated mullet.

Group		Final Mortality (%)	RPS (%)
Biofilm vaccine	*n* = 20	25	74
Whole-cell vaccine	*n* = 20	55	42
Chitosan particle	*n* = 20	70	26
PBS	*n* = 20	95	

**Table 3 vaccines-09-00844-t003:** Relative percent survival (RPS) in vaccinated mullet.

Group		Final Mortality (%)	RPS (%)
Biofilm vaccine	*n* = 25	20	77
Whole-cell vaccine	*n* = 25	72	18
Chitosan particle	*n* = 25	88	0
PBS	*n* = 25	88	

## Data Availability

The data presented in this study are available in this article.

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
