# Peer review of "Protective Efficacy of Novel Oral Biofilm Vaccines against Lactococcus garvieae Infection in Mullet, Mugil cephalus"

_vaccines, 2021, doi:10.3390/vaccines9080844_

Round 1

Reviewer 1 Report

This work is very interesting and significant due to the need to develop effective vaccines that protect farmed fish against bacterial infections.

However, entire text must be carefully revised for English writing and styling. Professional support is highly recommended.

Please reformulate following senteces/paragraphs, They are not clear and grammatically incorrect:
line 10 - immune cytokines (C3, TLR2, ...) - C3 is one of the complement protein, TLR2 is a receptor,
lines 40-41
lines 46-47
lines 61-62
lines 94-100
lines 114-119 - additionally - how the concentration of 1010cfu/ml was prepared to be accurate
line 125 - PBS group are oral administration 14 days vaccination
lines 162-165

Clarify what you mean:
FRP tank
line 84 - The L. garvieae was isolated from...on blood agar
2.6. Immunohistochemistry - 
please describe clearly the staining procedure
line 148 - sedation with 30ppm MS-222 of brackish water

RPS (line 14) - the abbreviation should be expanded in this line (where first time appears), not in line 198 
line 50 - its very hard to termed GALT as the organ
line 55-56 - unify the names of fishes - latin or english
Was the weight of the fish only 5 g? (line 72)
2.3.4. Preparation L. garvieae ... - use italics 

Above, I have listed only some of the comments and remarks.

The whole manuscript must be revised!

Reviewer 2 Report

7: grammar (pathogen cause)

8: grammar (administer), and this sentence needs to be reworded

15: based

16: grammar

Abstract overall: needs to be fixed for grammatical clarity

22: grammar

28: consists

39: this sentence is confusing

44: grammar

45: grammar

50: GALTs are not technically organs, just tissues

51: I get that you’re saying you need to include some part of biofilms in your vaccine, since the pathogen’s use of biofilms correlates to the worst outcomes of infection. I think you should end your first paragraph with the line “Therefore, bacterial biofilm…bacteria vaccine” at line 40. It emphasizes this point. Now in the next paragraph (lines 45-56), you need more info on using biofilms in your oral vaccines. Stating as you do in line 52: “biofilm with multiple immunogenicity can protect bacteria from gastric acid production”, this doesn’t seem like an adequate explanation of why adding biofilms is better for the vaccine. I get that it might work better, but expound on why, i.e. are there other antigens, is it causing increased inflammation, does it increase immune cell recruitment?

The other big problem is a contradiction between these two paragraphs. In line 33 you correctly point out that biofilm matrices can enable pathogen escape from the host immune system. But now you’re using biofilms almost as an adjuvant for your vaccine. I get that it works better, but that disparity is going to confuse people.

Your refs are also kindof old (I feel for you though since this is a mildly niche field). I would bring in some more updated stuff. You have Kahieshesfandiari‘s 2019 paper, but I would throw in some other papers, like Kaur 2021 or Ram 2019 or Kitiyodom 2021, among others

Introduction overall: lots of grammar fixes, and make sure biofilms as vaccine adjuvants are addressed

69 and 78: the italics (or not) is mildly confusing in consistency.

69:where did you get the fish

75 and 76: grammar

75: 2% of fish spleens doesn’t make sense. If you had 224 fish how many did you check?

78: where did this chitosan come from, and do you have any references for how to do this procedure?

84: Moribund black mullets is strange. Where did these black mullets come from? Are they in a different colony? Why not get the bacteria from a service? Did you sacrifice them or were they just floating dead in the tank?

90: italics for bacteria

91: this section needs to be cleaned up

114: again, is there a reference for this?

131: We don’t know your experiment numbers, call it something different, like challenge study

161: this section needs work

163: blood was drawn; you should write out tricaine mesylate instead of just the acronym

164: so you drew blood and let it sit in the fridge overnight? Is this normal SOP, is there a reference? Why not spin it down to get the serum? And how were the fish allowed to recover?

176: H2SO4 was added after 10 min, not for 10 min

192: you need refs in this section

204: what software did you use

Methods Overall: need a lot of work. Also, you didn’t talk about how you actually isolated organs like liver and kidney. You need to say that

Results section: What’s the point of showing the SEM stuff? It’s kindof hard to tell what you’re pointing out, and I don’t see how it’s really relevant to your story

It would help to quantitate your IHC, Fig. 3D is pretty weak

Why not have figures 4 and 5 merged?

Figure 4 says an n of 3…that’s really really low. I thought there were over 200 fish

Figure 5 y axis is spelled wrong

247: antibody is spelled wrong

Figure 6: not trying to be a jerk but your n, and the significance numbers you’re getting in these figures, seem off. The text styles are different too, and your letters for fig A and B are wonky

266: I would have liked some more immune-related genes, but these are ok. Maybe some markers of activation next time, or some interferons. Having some off-target genes for reference is an easy addition and helps sell your story (like a viral TLR would sell the TLR2)

286: starting, not staring

Figure 8: days post vaccination would be a more common x-axis. Was there any significance in the survival rates? Same with figure 10

Table 2 is ordered weird, keep it consistent with the other figures

Discussion: Honestly your discussion did a better job of explaining what you’re doing that the introduction

331: don’t change your ref style

385: you need to explain why this is important, not rehash the results again

Round 2

Reviewer 1 Report

The manuscript has been revised and improved. The changes that have been made allow for the publication of the paper in its present form.

Reviewer 2 Report

Much better, lots of good fixes. I would do one more pass to check for minor things like capitalization, etc., but overall a much improved paper